# Rapid Estimation of Soil Pb Concentration Based on Spectral Feature Screening and Multi-Strategy Spectral Fusion

**DOI:** 10.3390/s23187707

**Published:** 2023-09-06

**Authors:** Zhenlong Zhang, Zhe Wang, Ying Luo, Jiaqian Zhang, Duan Tian, Yongde Zhang

**Affiliations:** College of Environment and Resources, Southwest University of Science & Technology, Mianyang 621010, China

**Keywords:** soil Pb contamination, vis-NIR, XRF, feature selection, WOA, CARS, outer-product analysis, model averaging, multi-strategy fusion

## Abstract

Traditional methods for obtaining soil heavy metal content are expensive, inefficient, and limited in monitoring range. In order to meet the needs of soil environmental quality evaluation and health status assessment, visible near-infrared spectroscopy and XRF spectroscopy for monitoring heavy metal content in soil have attracted much attention, because of their rapid, nondestructive, economical, and environmentally friendly features. The use of either of these spectra alone cannot meet the accuracy requirements of traditional measurements, while the synergistic use of the two spectra can further improve the accuracy of monitoring heavy metal lead content in soil. Therefore, this study applied various spectral transformations and preprocessing to vis-NIR and XRF spectra; used the whale optimization algorithm (WOA) and competitive adaptive re-weighted sampling (CARS) algorithms to identify feature spectra; designed a combination variable model (CVM) based on multi-layer spectral data fusion, which improved the spectral preprocessing and spectral feature screening process to increase the efficiency of spectral fusion; and established a quantitative model for soil Pb concentration using partial least squares regression (PLSR). The estimation performance of three spectral fusion strategies, CVM, outer-product analysis (OPA), and Granger-Ramanathan averaging (GRA), was discussed. The results showed that the accuracy and efficiency of the CARS algorithm in the fused spectra estimation model were superior to those of the WOA algorithm, with an average coefficient of determination (R^2^) value of 0.9226 and an average root mean square error (RMSE) of 0.1984. The accuracy of the estimation models established, based on the different spectral types, to predict the Pb content of the soil was ranked as follows: the CVM model > the XRF spectral model > the vis-NIR spectral model. Within the CVM fusion strategy, the estimation model based on CARS and PLSR (CARS_D1+D2) performed the best, with R^2^ and RMSE values of 0.9546 and 0.2035, respectively. Among the three spectral fusion strategies, CVM had the highest accuracy, OPA had the smallest errors, and GRA showed a more balanced performance. This study provides technical means for on-site rapid estimation of Pb content based on multi-source spectral fusion and lays the foundation for subsequent research on dynamic, real-time, and large-scale quantitative monitoring of soil heavy metal pollution using high-spectral remote sensing images.

## 1. Introduction

Soil is an essential part of the human habitat. The abundance of metallic elements in the soil provides energy and resources for biological growth and human production and livelihood. However, extensive research has shown that high concentrations of Pb in the soil can cause varying degrees of damage to animals and plants in the area [1,2,3]. As a result, China has included Pb as one of the key pollutants to be focused on for prevention and control [4,5]. Priority is given to protecting soils that have suffered from Pb contamination, and soil Pb pollution monitoring is a prerequisite for effective governance and protection [6]. Therefore, it is crucial to accurately, rapidly, affordably, and environmentally assess the concentration of Pb in soil.

Traditional geochemical monitoring methods such as Inductively Coupled Plasma Optical Emission Spectroscopy (ICP-OES) and Atomic Absorption Spectroscopy (AAS) can accurately characterize the Pb content of soil [7,8]. However, these methods require pre-treatment of soil samples, leading to drawbacks such as lengthy testing times, complex experimental procedures, high experimental requirements, expensive analysis costs, and the potential for secondary pollution, making them inadequate for soil Pb monitoring needs [9,10,11]. In the case of Pb pollution in mining areas and their affected regions or river basins, traditional monitoring methods are insufficient for studying temporal and spatial variations. Additionally, the tracing and migration behavior of Pb are difficult to identify, which can hinder the speed of soil Pb pollution control and remediation [12,13]. Therefore, there is a need for convenient, accurate, and environmentally friendly spectroscopic techniques to determine Pb content. In recent years, X-ray fluorescence spectroscopy (XRF) and visible and near-infrared spectroscopy (vis-NIR) have been proven to be capable of estimating Pb content [14,15], and multifunctional mass spectrometry (MS) techniques [16] have demonstrated a greater potential for materials testing. Moreover, the advent of portable spectroscopic instruments has accelerated the acquisition of soil spectral information, enabling rapid on-site estimation of soil Pb content, reducing most of the analytical testing procedures, and improving the efficiency of soil heavy metal monitoring work [17].

Portable X-ray fluorescence spectroscopy (pXRF) is capable of providing on-site soil Pb content for a specific sampling point in a short amount of time. It is suitable for the real-time field assessment of soil Pb content and represents a low-cost and user-friendly method for monitoring soil heavy metal content [18,19]. Furthermore, when conducting soil Pb content tests in laboratory conditions, pXRF yields more stable data and provides more accurate data on the number of X-ray-excited electrons collected [20]. However, the results obtained with pXRF can still be uncertain due to factors such as soil physicochemical properties, element detection limits, interference from similar elements, and challenges in integrating with remote sensing technology, making it difficult to carry out large-scale spatial heavy metal pollution monitoring [21,22]. On the other hand, vis-NIR relies on the collection of visible and near-infrared spectral range light reflected from soil when illuminated with a halogen lamp. However, it contains limited information relevant to soil Pb, which limits its accuracy when directly measuring soil Pb content [23,24]. To address this limitation, the reliable estimation of soil heavy metal concentrations can be achieved based on the relationship between the content of soil organic matter, clay minerals, or iron oxides and vis-NIR spectra. Yet, this still requires pre-processing and quantitative testing analysis of intermediate media in laboratory conditions, making the testing process somewhat cumbersome [25,26,27]. However, vis-NIR offers greater convenience, as the spectral sensor can be mounted on portable devices or airborne platforms for remote sensing. Its use in watershed-scale soil heavy metal monitoring has been verified [28,29,30]. Vis-NIR boasts advantages such as non-invasiveness, low cost, real-time updates, and large spatial scale monitoring, making it highly applicable for watershed monitoring and holding immense potential for such applications.

Currently, using XRF or vis-NIR spectra alone does not meet the accuracy requirements for soil Pb content monitoring [23,31]. Factors such as the complexity of soil composition, spectral noise, sensor system errors, and the detection limit of Pb elements can interfere with the stability and accuracy of the estimation [11]. Compared to using a single spectral model, the fusion of XRF and vis-NIR spectra can enhance soil spectral information and improve the accuracy and efficiency of soil Pb concentration estimation models [32,33,34]. Among various spectral fusion strategies, outer product analysis (OPA) and Granger-Ramanathan averaging (GRA) are widely favored high-order fusion strategies and are commonly applied in the fusion process of XRF and vis-NIR spectra [35,36,37,38]. OPA involves the outer product analysis of the feature spectra of two types of spectra belonging to feature-level fusion and greatly increases the implicit information of the spectra [39,40]. On the other hand, GRA involves fitting the prediction results of the two types of spectra again, with spectra belonging to decision-level fusion, and the multiple fitting processes increase the accuracy of the estimation model [41]. In addition, there are few studies on model efficiency in soil heavy metal estimation models constructed through the strategy of fusing OPA and GRA [11]. Xu et al. [33] fused vis-NIR and XRF spectra using OPA and GRA, respectively, and successfully modeled the estimation of soil Cr. OPA gave the highest prediction accuracy with a Lin’s concordance correlation coefficient (LCCC) of 0.90. Inspired by the OPA and GRA spectral fusion strategies, we used partial least squares regression (PLSR) to establish a combined variable model (CVM) with simple principles and strong operability. We compared the accuracy and efficiency of the CVM, OPA, and GRA in the estimation models for soil Pb content.

Spectral fusion significantly increases the volume of spectral data, which puts pressure on the spectrum feature selection process [34,42]. Therefore, the application of evolutionary algorithms such as the Whale Optimization Algorithm (WOA) and Competitive Adaptive Reweighted Sampling (CARS) has been validated in the field of spectroscopy, as they can select important feature spectra from the original spectra [43,44,45]. The WOA has the advantages of fast convergence, ease of understanding, and simple debugging [46]. CARS is a feature variable selection method that combines Monte Carlo sampling with PLSR. After multiple computations, it selects the subset with the smallest root mean squared error of the cross-validated spectra (RMSECV) as the spectrum feature [47]. In addition, few studies have been conducted to construct soil heavy metal estimation models with fusion spectra using the WOA and CARS algorithms [45], and it is necessary to validate the performance of the two algorithms. Tan et al. [29] used CARS to screen the characteristic bands of the airborne vis-NIR spectra, and constructed a model with an R2 of 0.6 for soil Pb content estimation by combining multiple modeling methods. Simultaneously, spectral dimensionality reduction can retain the wavelength bands with high explanatory power for soil Pb, thus enhancing the efficiency and accuracy of spectrum feature selection [48,49]. The Pearson Correlation Coefficient (PCC) is an effective spectral dimensionality reduction method that can be used to select wavelength bands with high correlation to soil Pb content from the spectral data [50,51]. Therefore, using the PCC for preliminary screening of the fused spectra can effectively reduce the algorithm’s processing time.

The purpose of this study is as follows: (1) to compare the characteristics of the WOA and CARS algorithms in the spectrum feature selection process; (2) to utilize XRF and vis-NIR to estimate soil Pb content; (3) to compare the accuracy and efficiency of different spectral preprocessing methods in establishing soil Pb content estimation models; (4) to discuss the accuracy of soil Pb content estimation models established using three spectral fusion strategies—CVM, OPA, and GRA; and (5) to provide the technical means for accurately, rapidly, non-destructively, and cost-effectively estimating soil Pb content based on multi-source spectral fusion.

## 2. Materials and Methods

### 2.1. Study Area

The study area is located in Gejiu City, Honghe Hani and Yi Autonomous Prefecture, Yunnan Province, China, covering an approximate area of 6.5237 square kilometers. Its geographical coordinates are between 103.1900° and 103.2200° east longitude and 23.5000° and 23.5400° north latitude. The region has a subtropical plateau monsoon climate with abundant rainfall. Within the study area, there are 965,935 square meters of construction land and 2,609,041 square meters of farmland. Suspected sources of pollution include a waste residue heap from non-ferrous metal smelting, industrial wastewater, and waste residue generated during non-ferrous metal processing. Due to the combined effects of atmospheric deposition, rainfall, and irrigation canals, Pb pollution has also been detected in nearby farmland, and the Pb concentration has exceeded environmental standards, posing a threat to crop production and human health [52]. A total of 121 sampling points were arranged within a 2 km radius around the waste residue heap (see Figure 1). Given the direct impact of soil pollution on food security on agricultural land, a larger number of sampling points were set up in the farmland. Appendix A shows more information about the study area.

### 2.2. Material Collection

The material collection work includes soil sample collection and pre-treatment, the use of chemical reagents, and the operation of experimental instruments and equipment. Throughout the experiments, we strictly adhered to relevant specifications to ensure the rigor of the research data.

#### 2.2.1. Soil Sample Collection

According to the Technical Specification for Soil Environmental Monitoring (TSSEM) [53], a total of 121 standard soil samples were collected, and the sampling point locations were recorded using GPS as shown in Figure 1. A few sampling points were located in shrubs around the waste residue heap, while most of them were situated in farmland surrounding the waste residue heap, with a minimum distance of 20 m from the road. During soil sample collection, we took precautions to avoid any contact between the soil and metal objects. Firstly, we removed surface materials like branches and weeds from the soil. Then, we used plastic or wooden shovels to collect the topsoil (0–5 cm). Five samples weighing over 200 g each were collected within a 10 m area using the “X-shaped sampling method”. These soil samples were mixed to form one composite sample. After removing stones, plant roots, and other impurities, the mixed soil sample weighed approximately 1 kg and was sealed in polyethylene plastic bags. The dried soil samples were ground in non-metallic grinding bowls. A 5 g portion of soil was sieved through a 100-mesh nylon sieve (0.1500 mm) for chemical analysis, while a 150 g portion of soil was sieved through a 60-mesh nylon sieve (0.4200 mm) for spectral measurements. Soil properties and soil maps are shown in Appendix A and Appendix A.

#### 2.2.2. Chemical Analysis

Firstly, 0.1000 g of the sample was weighed and placed in a Teflon digestion tank. Then, the samples were soaked in 5 mL of nitric acid for 0.5 h to remove the organic matter, followed by 2 mL of hydrofluoric acid and 1 mL of perchloric acid. Finally, the digestion tank was placed in a graphite digester and digested at 180 °C for 4 h. The total concentrations of Cd in the solutions were measured using an inductively coupled plasma emission spectrometer (ICP800DV, TMO, Waltham, MA, USA) (argon gas source valve pressure reduced at about 550 kPa, circulating water pressure indicated between 50 and 310 kPa) for 48 samples per batch, including two blank controls [54]. All experimental samples and the control standard for total Pb in soil were processed at the same time, and the ratio of the measured total Pb in the samples to the standard content was between 90% and 95%, indicating that the results were within the standard range.

#### 2.2.3. Measurement of vis-NIR and XRF Spectra

The vis-NIR spectra were measured in a darkroom under laboratory conditions using the Spectral Evolution PSR-2500 portable spectrometer (operating instructions are available on the Spectral Evolution website (https://spectralevolution.com/products/software/ (accessed on 3 May 2023))) manufactured by Spectral Evolution. The spectral range covered 350–2500 nm. The soil samples were placed in culture dishes with a diameter of 10 cm and a depth of 1 cm. Before measurement, a 100-watt halogen lamp was set as the sole light source, and the spectrometer was preheated for 30 min. Prior to each measurement, the spectrometer was optimized using a calibration whiteboard. The probe’s viewing angle was set at 15°, and the incident angle of the light source was set at 30°. The distance from the light source to the center of the soil surface was 50 cm, and the probe was maintained at a distance of 15 cm from the soil surface. To minimize measurement anomalies and instrument errors, a plastic ruler was used to level the soil surface. The container was then divided into three directions with angles of 120°, and five spectra were collected in each direction. A total of 15 spectra were averaged to represent one spectrum for the sample.

The XRF spectra were measured using the Niton XL3t 950 X-ray Fluorescence Spectrometer, manufactured by Thermo Fisher Scientific, Waltham, MA, USA. The spectrometer was connected to the computer via a data cable and the spectral data exported to the computer using the NDT program, the manual for which is available on the Thermo Fisher Scientific website (https://www.thermofisher.cn/order/catalog/product/10131166?SID=srch-srp-10131166 (accessed on 4 May 2023)). The soil samples were further ground through a 200-mesh nylon sieve (0.0740 mm) and placed in sample cups. The samples were then compacted to create a flat surface and covered with a mylar film. The sample cups were placed on the instrument’s detection platform for testing, and the XRF spectroscopy measurement was set to “soil mode”. Each sample was scanned for 90 s, and three scans were performed, with the average spectrum of the three scans taken as the final result.

### 2.3. Spectral Data Preprocessing

All data preprocessing and estimation model construction in this study were implemented using the Python 3.10 programming language in the PyCharm Community Edition 2023.1.1 software.

#### 2.3.1. Spectral Organization and Denoising

Firstly, the vis-NIR and XRF spectral data were organized in Microsoft Excel 2016 software for easy batch access by the program. To reduce the influence of edge noise and low-energy values, we selected the numerical values in the 400–2444.9 nm wavelength range for vis-NIR spectra and the 1.05–36 keV wavelength range for XRF spectra. Then, the spectra were further compressed using Daubechies 8 wavelet filtering to reduce noise [55,56]. Subsequently, spectral transformations were applied to enhance the spectral signals [57,58]. The employed spectral transformation methods included Standard Normal Variate (SNV), Multiplicative Scatter Correction (MSC), First Order Derivative (D1), Second Order Derivative (D2), Continuum Removal (CR), and Logarithm Reciprocal (CL). Further, the Savitzky-Golay (SG) algorithm with a window size of 12 and a polynomial order of 2 [59] was used to reduce noise and enhance signals. The PCC was then used to select the wavelength bands in the vis-NIR and XRF spectra that exhibited significant correlations with soil Pb content. The number of wavelength bands in the vis-NIR and XRF spectra was reduced to 400 and 600, respectively, to minimize data redundancy. Finally, all data was standardized to have a mean of 0 and a standard deviation of 1. The preprocessing steps are shown in Appendix A.

#### 2.3.2. Spectral Feature Selection

The WOA [60] is a population-based intelligent optimization algorithm that imitates the foraging behavior of humpback whales in nature to achieve optimization goals. It has the advantages of simple principles and fewer parameter settings. The training set is divided into a model training set and a model validation set after being input into the WOA. The objective function is set as the root mean square error (RMSE) of the model validation set, which makes the selected variables more representative. The iteration number, the number of whales, and the threshold for binary encoding are set as 1000, 50, and 0.3, respectively.

CARS [47] utilizes an adaptive sampling approach to retain spectral bands with relatively larger absolute coefficients in the PLSR model. Then, the Monte Carlo cross-validation method is used to model each subset of wavelength variables, and the optimal subset is selected based on the RMSE in the cross-validation. The iteration number, maximum number of principal components, and number of cross-validations are set at 100, 20, and 10, respectively.

The flowcharts of the WOA and CARS, the fitness curve of the WOA, the RMSECV curve of the CARS, the iteration curve of the CARS, and the position of selected feature spectra are shown in Appendix A.

### 2.4. Soil Pb Concentration Estimation Model Construction

We used the Kennard-Stone (KS) algorithm [61] to divide the data into training and validation sets in a 4:1 ratio, and then built the soil Pb concentration estimation model using PLSR.

#### 2.4.1. PLSR

PLSR is a novel multivariate statistical analysis method that combines the advantages of the PCC, principal component analysis (PCA), and linear regression models, making it more conducive to distinguishing spectral information from noise [62]. Compared to traditional linear models, PLSR features data dimension reduction and information synthesis selection techniques, enabling the modeling of variables with multiple correlations and reducing the correlations between variables [63]. The PLSRegression function is from the Scikit-Learn package with the feature dimension of PCA set to 8 and other parameters defaulted.

#### 2.4.2. Model Evaluation

The accuracy of the model is evaluated using the coefficient of determination (R^2^) and the RMSE. R^2^ reflects the stability of the model, where the closer R^2^ is to 1, the better the model’s fitting effect. A smaller RMSE indicates a better predictive ability of the model. The formulas for calculating R^2^ and RMSE are as follows:(1)$$R^{2}=\sum_{i=1}^{n}{\left(\hat{y_{i}}-\bar{y_{i}}\right)}^{2}/\sum_{i=1}^{n}{\left(y_{i}-\bar{y_{i}}\right)}^{2}$$
(2)$$\mathrm{RMSE}=\sqrt{\frac{1}{n}\sum_{i=1}^{n}{\left(\hat{y_{i}}-\bar{y_{i}}\right)}^{2}}$$
where $\bar{y}$ is the mean value of the sample observations, $\hat{y}$ is the predicted value of the sample, and *n* is the number of samples to be verified.

In order to enable the estimation model to quickly estimate soil Pb content in the field, we recorded the computation time of the program and further compared the efficiency of different soil Pb content estimation models.

### 2.5. Spectral Fusion

Spectral fusion consists of two stages: dominant model selection and feature-level concatenation. By combining two types of spectra (vis-NIR and XRF), six spectral transformation methods (SNV, MSC, D1, D2, CR, and CL), and two spectral feature selection algorithms (WOA and CARS), we obtained a total of 24 soil Pb concentration estimation models. These models were then categorized into four classes based on spectral types and spectral feature selection algorithms. The models with a higher accuracy (R^2^ > 0.5 for vis-NIR spectra and R^2^ > 0.8 for XRF spectra) were selected from each class. Finally, the selected feature spectra corresponding to each model were concatenated according to different spectral types and the same spectral feature selection algorithm. The concatenated spectra were further categorized into two fusion types: WOA_X+Y and CARS_X+Y (X represents the transformation method for vis-NIR spectra, and Y represents the transformation method for XRF spectra). The fusion spectra were subjected to spectral feature selection again using the WOA or CARS algorithms, and the CVM was established using PLSR.

To compare the accuracy and efficiency of the CVM with other spectral fusion models such as GRA and OPA, we implemented GRA and OPA models based on the WOA and CARS algorithms, respectively. The GRA model was established using PLSR based on the single-spectrum model. The OPA model required reducing both types of spectra to a unified dimension using the PCC. To avoid excessive computation time, the dimension of vis-NIR and XRF spectra was set to 100 wavelength bands. The technical workflow adopted in this study is shown in Figure 2.

## 3. Results

### 3.1. Descriptive Statistical Analysis of Soil Pb Content

In order to comply with the “Technical Specifications for Soil Environmental Monitoring in China” (HJ/T 166-2004), we measured the Pb content of 121 soil samples in the study area. To better represent the actual situation of Pb content in agricultural soil within the influence area of the metallurgical slag site, we kept highly polluted sampling points. As shown in Figure 3, the skewness value (3.74) and kurtosis value (20.27) indicate that the distribution of Pb content follows a positively skewed distribution. Normally, the frequency of element content in soil under natural background conditions conforms to a normal or log-normal distribution. However, the Pb element content in this area exceeds the natural background value, indicating soil pollution by Pb.

In the agricultural soil within the influence area of the metallurgical slag site, 112 sampling points (92.56%) exceed the screening value of the “Guideline for the Risk Control of Soil Pollution in Agricultural Land” (GB15618-2018) [54], and 8 sampling points (6.62%) exceed the control value, indicating severe Pb pollution in the study area. The coefficient of variation (CV) reflects the degree of dispersion of sample data and is one of the indicators reflecting the distribution of data. When the coefficient of variation is ≥1, it indicates strong variability; when 0.1 < coefficient of variation < 1, it indicates moderate variability; and when the coefficient of variation ≤0.1, it indicates weak variability. The CV of Pb content in the study area is 0.76, indicating that the soil Pb pollution in the study area is influenced by human activity. Before constructing the estimation model, it is necessary to partition the dataset into training and validation data. According to the KS algorithm, we divided the 121 sampling points into 97 training points and 24 validation points. The distribution of the training and validation data sets is similar to that of the overall data set, indicating that the data partitioning is representative. When the transformation methods of vis-NIR and XRF spectra are different, the Euclidean distance in the KS algorithm will change, and the distribution of the training and validation sets will also change, but this does not affect their similar distribution patterns.

### 3.2. Soil Pb Content Estimation Models Based on a Single Spectrum

From Table 1, it can be observed that, for the vis-NIR spectra with the same spectral transformation method, the accuracy and computation time of the soil Pb content estimation model based on the WOA spectrum feature selection algorithm (WOA model) are generally better than those of the model constructed based on the CARS spectrum feature selection algorithm and the PLSR method (CARS model). Among them, the models constructed based on five methods: WOA_D1, WOA_D2, WOA_CL, CARS_D1, and CARS_D2, show better accuracy and efficiency with all R^2^ exceeding 0.5, which provides a set of advantageous vis-NIR spectra-based soil Pb content estimation models for subsequent spectral fusion model construction.

From Table 1, it can also be observed that, for the XRF spectra with the same spectral transformation method, the overall accuracy of the CARS models is better than that of the WOA models, while the computation time of the WOA models and the CARS models is similar. Among them, the models constructed based on five methods: CARS_D1, CARS_D2, CARS_CR, WOA_D1, and WOA_D2, show better accuracy and efficiency with all R^2^ exceeding 0.8, which provides a set of advantageous XRF spectra-based soil Pb content estimation models for subsequent spectral fusion model construction.

Among the soil Pb content estimation models constructed based on different spectral transformation methods and spectrum feature selection algorithms for the vis-NIR spectra, the model constructed using the WOA_D1 method shows the highest R^2^ and the smallest RMSE, which are 0.6881 and 0.2319, respectively. For the XRF spectra, the model constructed using the CARS_D2 method shows the highest R^2^ and the smallest RMSE, which are 0.9244 and 0.1653, respectively. At the same time, the accuracy of the soil Pb content estimation models based on XRF spectra is generally better than that of those based on vis-NIR spectra.

### 3.3. Soil Pb Content Estimation Models Based on Spectral Fusion

From Table 2, it can be observed that the R^2^ of the soil Pb content estimation models constructed based on the fusion of vis-NIR and XRF spectra are all above 0.8, indicating high accuracy of the models constructed based on spectral fusion. Among them, the accuracy of the models using the CARS method is better than that of the models using the WOA method. The average R^2^ value (0.9226) and RMSE value (0.1984) of the CARS models are higher and better than those of the WOA models (R^2^: 0.8300, RMSE: 0.3770). Additionally, the average computation time (428.5000 s) of the CARS models is shorter than that of the WOA models (528.8333 s). Further, the soil Pb content estimation model based on the CARS_D1+D2 fusion method shows relatively superior accuracy and efficiency, with a R^2^ of 0.9546, a RMSE of 0.2035, and a computation time of 468 s.

We created scatter plots with the measured soil Pb content on the *X*-axis, the model-estimated values on the *Y*-axis, and the absolute errors between the measured and estimated values on the *Z*-axis. These scatter plots depict the comparison between the actual measurements and the predictions made using the 12 soil Pb content estimation models constructed using the CVM (Combination of Variables Models) strategy. The scatter plots, including Figure 4, Appendix A and Appendix A, illustrate the performance of different spectral feature selection algorithms within the CVM strategy. In these scatter plots, the colored bands on the ground surface represent the projection of the three-dimensional graph onto the plane. The color bands vary from purple to red, indicating an increasing trend in absolute errors. The intersection of the three-dimensional graph with the ground plane (the middle line of the purple region) represents the 1:1 line, where the measured and estimated values are equal. The closer the scatter points are to the 1:1 line, the smaller the absolute errors, and the better the model’s accuracy. The trend of the scatter points in relation to the 1:1 line reflects the model’s fitting accuracy.

Figure 4 specifically displays the scatter plot of the best-performing models, WOA_D2+D1 and CARS_D1+D2, based on the CVM fusion strategy. It is evident that the scatter points of the WOA_D2+D1 model are more scattered and further away from the 1:1 line, with some absolute errors exceeding the range of the purple region. On the other hand, the scatter points of the CARS_D1+D2 model are more concentrated around the 1:1 line, indicating that the estimated values are closer to the actual measurements, resulting in smaller absolute errors. Furthermore, compared to the WOA_D2+D1 model, the CARS_D1+D2 model shows a smaller deviation from the 1:1 line, indicating a higher level of fitting and better accuracy for the estimation model. Based on these observations, the CARS_D1+D2 model within the CVM strategy is identified as the optimal estimation model for soil heavy metal Pb content.

### 3.4. Contrastive Analysis of Estimation Model Accuracy and Efficiency

Figure 5 presents a comparative analysis of accuracy and computation time for three categories of the 36 soil Pb content estimation models constructed using single spectrum and fused spectra, represented as violin plots (the outer part shows data kernel density contours, and the inner part shows data box plots). From Figure 5, it can be observed that, for the vis-NIR spectra-based models, the R^2^ box plot of the WOA models is superior to that of the CARS models, with R^2^ values around 0.5 for the WOA models and in a range of 0.1–0.4 for the CARS models. However, in the case of XRF spectra and CVM-fused spectra, the CARS models outperform the WOA models with higher R^2^ values and lower RMSE values, and the computation time for CARS models is slightly advantageous.

Regarding the single spectrum, the R^2^ and RMSE contours of the WOA models exhibit multi-modal characteristics, indicating instability and difficulty in controlling the accuracy of the estimation models within an acceptable range. Conversely, in the CARS models, both the single and fused spectra exhibit unimodal R^2^ and RMSE contours, suggesting a more concentrated estimation accuracy. The peak positions of R^2^ kernel density contours for vis-NIR spectra, XRF spectra, and CVM-fused spectra gradually shift upwards, with CVM-fused spectra having a peak close to 1, indicating higher accuracy. Similarly, the peak positions of RMSE kernel density contours for the three spectra types gradually shift downward, with CVM-fused spectra having a peak close to 0.1, indicating lower error. This suggests that the CVM-fused spectra-based soil Pb content estimation model has higher accuracy. Moreover, the R^2^ and RMSE contours for vis-NIR spectra and XRF spectra are wider and flatter (lower peak, larger width), while the CVM-fused spectra-based contour is higher and narrower (higher peak, smaller width), indicating that the CARS models based on CVM-fused spectra exhibit lower variability and higher stability.

In conclusion, compared to single spectrum-based soil Pb content estimation models, the fused spectra-based models demonstrate higher accuracy and better stability. Additionally, the CARS algorithm is more suitable for feature selection in fused spectra, outperforming the WOA algorithm.

## 4. Discussion

### 4.1. Spectral Feature Selection Algorithms

Spectral data often have high dimensions, and previous studies have shown that not all spectral information positively contributes to the accuracy of soil Pb content estimation models [42,51]. Therefore, it is necessary to select spectral bands to improve the efficiency and accuracy of estimation models. In this study, we used the WOA and CARS algorithms to select feature spectra from vis-NIR and XRF spectra, respectively, and established soil Pb content estimation models based on the PLSR method. The results in Table 1 show that the choice of algorithm significantly influences the accuracy and efficiency of the estimation models, which is consistent with the findings of Gholizadeh et al., who used a univariate filter (UF) and genetic algorithm (GA) for spectral feature selection [31]. Similar phenomena have been observed in other studies as well [64,65]. Therefore, spectral feature selection is crucial in establishing the relationship between spectral data and soil heavy metal content.

Among numerous spectral feature selection algorithms, the WOA algorithm, which simulates the foraging behavior of animals, and the CARS algorithm, which simulates the adaptation of organisms to environmental changes, are representative approaches [45]. Although the application of the WOA algorithm in estimating soil Pb content is relatively limited, our study shows that in the vis-NIR spectrum, the WOA model achieves the highest R^2^ of 0.6881, outperforming the CARS model (Table 1 and Figure 5). Bian et al. used the WOA algorithm to select feature spectra from near-infrared spectra and combined them with PLSR to quantitatively predict sunflower oil in mixed oils, achieving a high prediction R^2^ of 0.9635 [66]. Thus, the WOA algorithm exhibits significant advantages in near-infrared spectral applications. Furthermore, Tan et al. employed the CARS algorithm to select feature bands from airborne vis-NIR spectra and constructed soil Pb content estimation models using various modeling methods, with the highest R^2^ of the validation set reaching 0.60 [29]. This is similar to our highest R^2^ value (0.6668) obtained by applying the CARS algorithm for spectral feature selection in vis-NIR spectra combined with PLSR (Table 1). Additionally, regardless of the spectral feature selection algorithm used, an estimation using vis-NIR spectra for Pb content is slightly inferior to that using XRF spectra (Table 1 and Figure 5), which is consistent with previous research on soil Pb element determination using XRF and vis-NIR spectra [67].

### 4.2. Spectral Fusion Strategies

Spectral fusion can be categorized into three stages: data-level fusion, feature-level fusion, and decision-level fusion [34]. Previous studies have shown that estimation models constructed using feature-level fusion, represented by OPA, and decision-level fusion, represented by GRA, outperform models based on data-level fusion [32,68,69,70]. OPA can effectively utilize the different properties and complementary information of XRF and vis-NIR spectra, thereby improving the prediction accuracy of soil heavy metal estimation models [33]. Our study also validated this conclusion (Table 1 and Table 2), and the performance of the soil Pb content estimation model using the OPA strategy is consistent with previous research [67] (Figure 6). Additionally, the GRA strategy is widely favored because it only requires the addition of a simple linear regression model. Thus, we applied the GRA strategy to fuse XRF and vis-NIR spectra, and the results were consistent with previous studies [70] (Figure 6). Therefore, both OPA and GRA strategies provide effective methods for using XRF and vis-NIR spectra in soil Pb content estimation.

Inspired by the OPA and GRA spectral fusion strategies, we designed a CVM based on a two-layer (feature-level and decision-level) fusion strategy, and compared the accuracy and computational time of CVM, OPA, and GRA in the soil Pb content estimation model. Figure 6 shows the comparison of accuracy and efficiency of the soil Pb concentration estimation models under different spectral feature selection algorithms and spectral fusion strategies using vis-NIR spectra in the D1 transformation and XRF spectra in the D2 transformation. It can be observed that the CVM (CARS) estimation model exhibits the highest R^2^ (0.9643), the lowest RMSE (0.1842), and the shortest computational time (149 s). Compared to the strategy of solely using feature-level concatenation for intermediate spectral fusion in previous studies [67], our CVM strategy shows significant improvement in accuracy, and the R^2^ of the estimation model is slightly superior to advanced fusion strategies (OPA and GRA) (Figure 6). Additionally, we found that after using OPA for the feature-level fusion of spectral data, the number of bands sharply increases, leading to longer running times. Nevertheless, the OPA (CARS) estimation model shows the smallest RMSE (0.1661) and a relatively high R^2^ (0.9515), which is consistent with Xu’s research findings [67]. On the other hand, the estimation model constructed after using GRA for the decision-level fusion of spectral data demonstrates a more balanced accuracy, similar to the results obtained by Shrestha et al. [71], but it requires a longer computational time (average computational time = 331). Therefore, the CVM, OPA, and GRA spectral fusion strategies each have their advantages in providing accurate, efficient, and stable methods for soil Pb content estimation models.

In summary, this study performed fusion on the vis-NIR and XRF spectra that were pre-screened by the PCC. The WOA and CARS algorithms were employed to identify feature spectra. Among them, the CARS spectral feature selection algorithm, in combination with the PLSR method, constructed the optimal estimation model for soil Pb content (CARS_D1+D2), demonstrating excellent estimation accuracy and stability. These findings provide technical means for on-site rapid estimation of soil Pb content based on multisource spectral fusion, enriching the technical methods for monitoring soil Pb concentration using spectral techniques [11]. Furthermore, they lay the foundation for subsequent research on the dynamic, real-time, and large-scale quantitative monitoring of soil heavy metal pollution based on hyperspectral remote sensing images [30,72].

## 5. Conclusions

In conclusion, the study successfully implemented the fusion of vis-NIR and XRF spectra using the CVM, OPA, and GRA fusion strategies. Spectral feature selection was performed on the single spectrum and fused spectra using the WOA and CARS algorithms, respectively. Soil Pb content estimation models were established based on both the single spectrum and fused spectra using the PLSR method. The comprehensive efficiency (accuracy and computation time) ranking of the estimation models based on different spectral types was as follows: fused spectra model > XRF spectra model > vis-NIR spectra model. The comprehensive efficiency ranking based on different spectral feature selection algorithms was: CARS algorithm > WOA algorithm. Lastly, the comprehensive efficiency ranking based on different fusion strategies was: CVM strategy > OPA strategy > GRA strategy. Importantly, among all the estimation models, the CARS_D1+D2 fused model exhibited the highest R^2^, a smaller RMSE, and better stability, making it more suitable for dynamic, real-time, and quantitative monitoring of soil heavy metal pollution. Future work should focus on constructing estimation models that can be used for on-site rapid and accurate estimation of soil Pb content, which is crucial for addressing the dynamic monitoring of soil pollution and agricultural product safety, as well as the safe utilization of cultivated land.

## Figures and Tables

**Figure 1 sensors-23-07707-f001:**
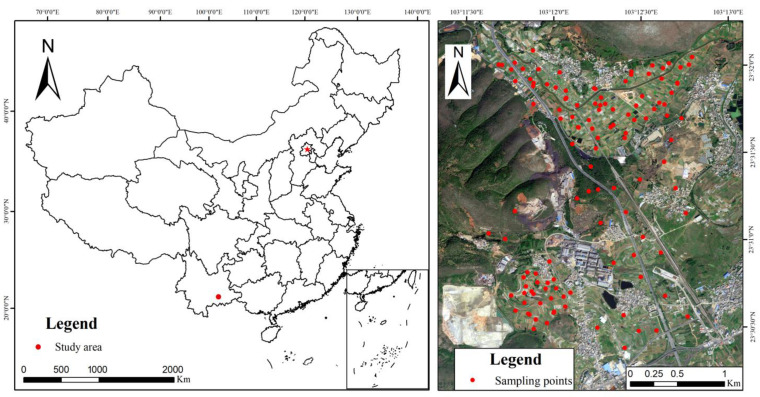
Study area. The five-pointed star represents the location of the capital of China.

**Figure 2 sensors-23-07707-f002:**
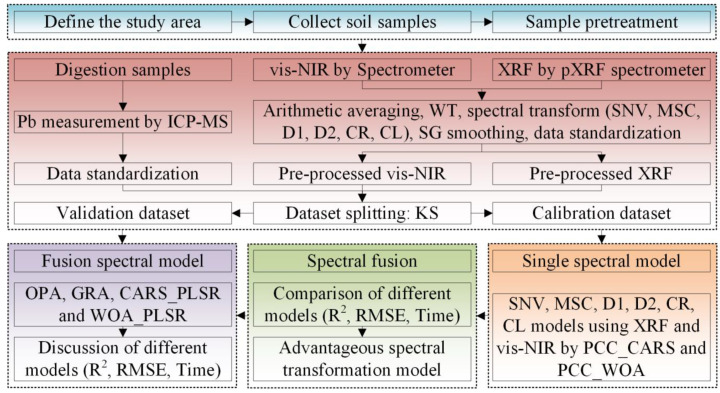
Technical flow chart of soil Pb concentration estimation model construction.

**Figure 3 sensors-23-07707-f003:**
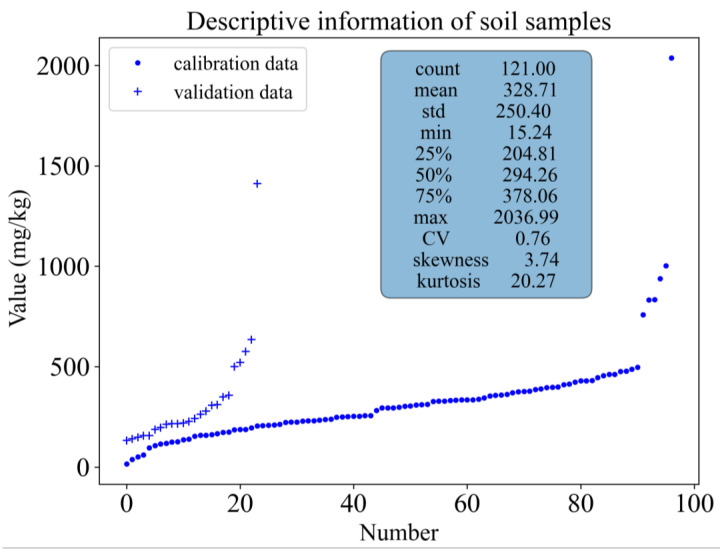
Descriptive statistics of Pb content and division of data sets.

**Figure 4 sensors-23-07707-f004:**
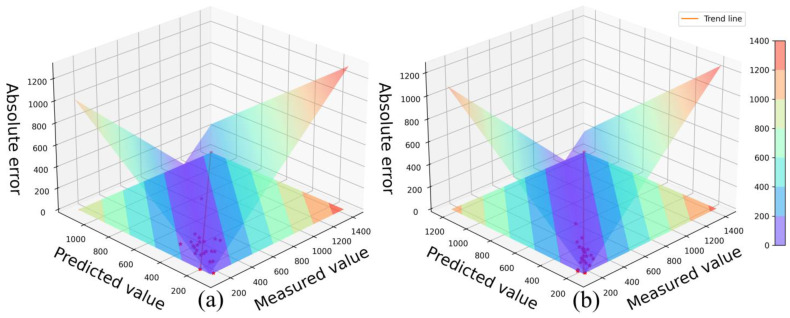
The scatter diagram of the measured and estimated values of the soil Pb content estimation model constructed using the CVM strategy ((**a**) is WOA_D2+D1, (**b**) is CARS_D1+D2).

**Figure 5 sensors-23-07707-f005:**
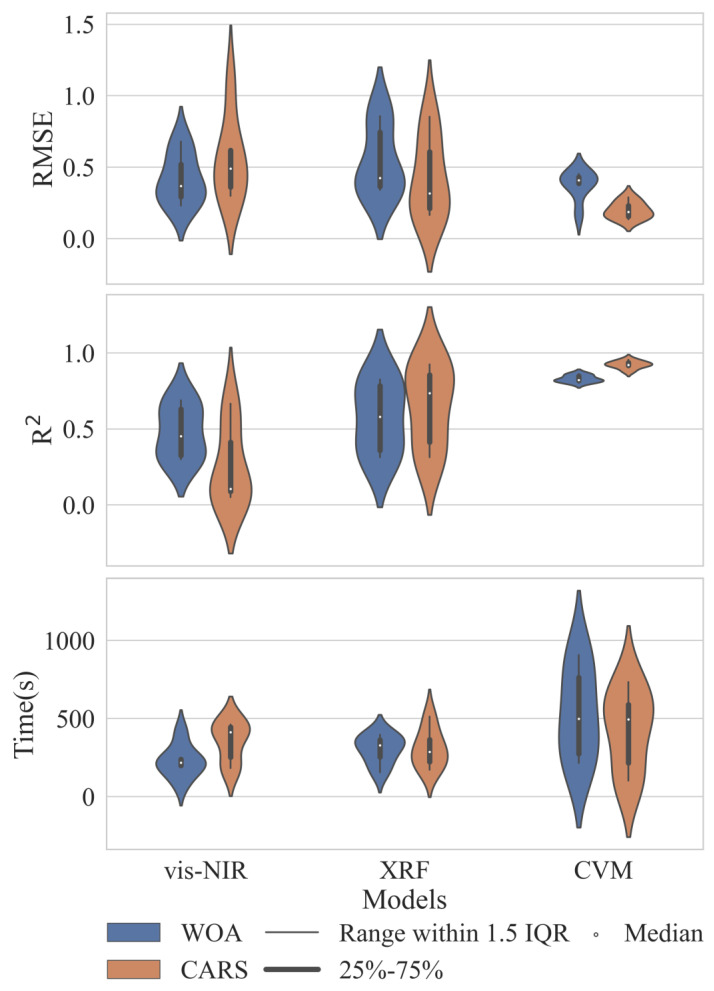
Violin plot comparing single and fused spectral estimation models’ accuracy and efficiency.

**Figure 6 sensors-23-07707-f006:**
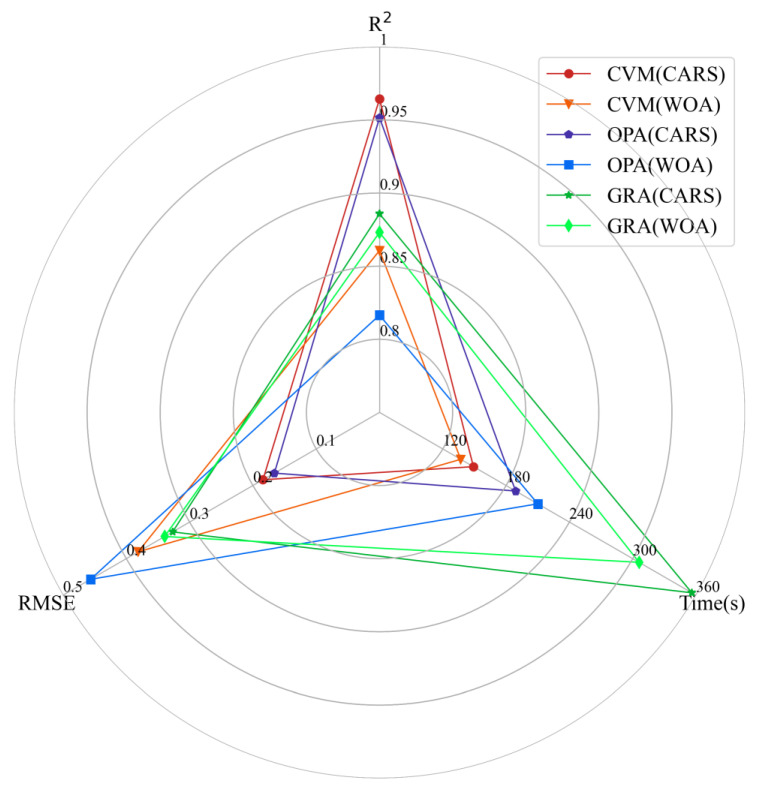
Accuracy and efficiency radar charts of different fusion strategies.

**Table 1 sensors-23-07707-t001:** Statistical table of accuracy and efficiency of single spectrum estimation model for soil Pb content.

Spectrum	Method	RMSE	R^2^	Time(s)
vis-NIR	WOA_SNV	0.5692	0.3592	410
	WOA_MSC	0.3643	0.3129	229
	WOA_D1	0.2319	0.6881	206
	WOA_D2	0.2694	0.6589	197
	WOA_CR	0.3746	0.3011	89
	WOA_CL	0.6800	0.5466	242
	CARS_SNV	1.0865	0.0511	416
	CARS_MSC	0.3955	0.0856	200
	CARS_D1	0.3482	0.5115	183
	CARS_D2	0.2987	0.6668	463
	CARS_CR	0.6306	0.0978	455
	CARS_CL	0.5845	0.1155	410
XRF	WOA_SNV	0.8508	0.3139	238
	WOA_MSC	0.8576	0.3198	397
	WOA_D1	0.4342	0.8260	302
	WOA_D2	0.4152	0.8169	155
	WOA_CR	0.3479	0.6835	366
	WOA_CL	0.3405	0.4769	357
	CARS_SNV	0.8535	0.3133	513
	CARS_MSC	0.6711	0.3456	310
	CARS_D1	0.2143	0.8556	209
	CARS_D2	0.1653	0.9244	265
	CARS_CR	0.2102	0.8531	383
	CARS_CL	0.4205	0.6167	172

**Table 2 sensors-23-07707-t002:** Statistical table of accuracy and efficiency of fusion spectra estimation model of soil Pb content.

Method	RMSE	R^2^	Time(s)
WOA_D1+D1	0.1729	0.8552	560
WOA_D1+D2	0.4052	0.8233	221
WOA_D2+D1	0.3813	0.8607	437
WOA_D2+D2	0.4118	0.8189	832
WOA_CL+D1	0.4523	0.8185	216
WOA_CL+D2	0.4384	0.8031	907
CARS_D1+D1	0.2883	0.9180	132
CARS_D1+D2	0.2035	0.9546	468
CARS_D1+CR	0.1515	0.9236	522
CARS_D2+D1	0.2414	0.9396	613
CARS_D2+D2	0.1710	0.9188	734
CARS_D2+CR	0.1347	0.8810	102

## Data Availability

The data and materials used and analyzed in the current study are available from the corresponding author on reasonable request.

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
