# Peer review of "Rapid Estimation of Soil Pb Concentration Based on Spectral Feature Screening and Multi-Strategy Spectral Fusion"

_sensors, 2023, doi:10.3390/s23187707_

Round 1
Reviewer 1 Report
In this paper, the authors successfully implemented the fusion of vis-NIR and XRF spectra using the CVM, OPA, and GRA fusion strategies. Feature spectral selection was performed on the single spectra and fused spectra using the WOA and CARS algorithms, respectively. Soil Pb content estimation models were established based on both single spectra and fused spectra using the PLSR method.The interesting points can be seen. However, the organization and preparing of this manuscript can be improved. If this manuscript is considered to be published, unless,the some revision has been done.
(1)English in this paper needs improvement, which can make this paper more like a journal paper.
(2) Please polish the abstract. Please check the logic of abstract. Please add sentences to explain the meaning, the main points, the improvement and the promising application of the study. Plenty of detail data have given, however, in abstract, important procedures and results should be mentioned in simple manner. Please focus on the main points and the improvement of the study.
(3) Please highlight the advance of the study in Introduction. Please explain the development and creative work. The literature review should be carefully considered.
(4)The flow chart of WOA and CARS algorithms can be considered to given.
(5) The more information about PLSR should be given,
(6) The configuration of the measurement should be given.
(7) What are the potential limitations or challenges in implementing the proposed results in real-world applications?
(8) Can the proposed results be further optimized to expand its capabilities beyond specific case?
(9) The figures in the Supplementary Materials is not clear,and please replote them.
(10)So far, many types of material detection technology have been reported. The significant improvement in concept in those fields can be considered to add in the introduction to enrich the background of such a topic. For instance,
*Annalen der PhysiK, 2023, vol.535, no.5, pp.2300030. DOI: 10.1002/andp.202300030
It can be improved.
Reviewer 2 Report
some comments are presented in the attached file

Reviewer 3 Report
The manuscript is very clearly written and the parts are in harmony with each other.
The authors aimed to provide technical means for rapid in situ estimation of Pb content in soil based
multisource spectral fusion. The intention is excellent to add technical methods for monitoring the
Pb concentration in soil using spectral techniques. These are studies that can have a real impact by providing real-time, large-scale estimation of heavy metal soil pollution based on hyperspectral remote sensing images.
I find points that the authors should address:
line 141 - a large number of sampling points is indicated. Did the sampling occur simultaneously on the same day and was the same weather condition ensured? In particular, is it ensured that there were no rain events that might have contributed to the variability of the sample?
line 302-303 The causes of a high value of coefficient of variability can be many. If the authors are certain of this relationship between coefficient of variability and the influence of human activities, then a literature source confirming this should be given. Otherwise, such a high coefficient of variability should call for more attention to the sources of variability. This means that one should assess whether the variability decreases or increases by dividing the points into different zones.
For example, it was indicated that some points came from areas with shrubs, others from agricultural areas or even other areas with different characteristics. Or one should subdivide the points by distance categories from areas that could be the cause of Pb release.
line 482 - Figure 6- Very illustrative, but uses colors that might not be distinguished by color-blind people. I suggest adding different symbols for each strategy represented in addition to the colors.
Round 2
Reviewer 1 Report
The authors have revised the manuscript based on the comments, and I think the manuscript can be published with such a form.
The English can be improved.
Reviewer 3 Report
Good morning, I read the authors' responses according to the suggestions that had been sent. I find that the authors have answered and resolved all the reports that had been highlighted.